# Analysis and Comparison of Rapid Methods for the Determination of Ochratoxin a Levels in Organs and Body Fluids Obtained from Exposed Mice

**DOI:** 10.3390/toxins14090634

**Published:** 2022-09-13

**Authors:** Zsuzsanna Szőke, Bianka Babarczi, Miklós Mézes, István Lakatos, Miklós Poór, Eszter Fliszár-Nyúl, Miklós Oldal, Árpád Czéh, Kornélia Bodó, György Nagyéri, Szilamér Ferenczi

**Affiliations:** 1Institute of Genetics and Biotechnology, Department of Animal Biotechnology, Hungarian University of Agriculture and Life Sciences, Agribiotechnology and Precision Breeding for Food Security National Laboratory, Szent-Györgyi Albert Street 4, H-2100 Gödöllő, Hungary; 2Institute of Physiology and Nutrition, Department of Feed Safety, Hungarian University of Agriculture and Life Sciences, Páter K. Street 1, H-2100 Gödöllő, Hungary; 3Department of Pharmacology, Faculty of Pharmacy, University of Pécs, Szigeti Street 12, H-7624 Pécs, Hungary; 4Food Biotechnology Research Group, János Szentágothai Research Centre, University of Pécs, Ifjúság Street 20, H-7624 Pécs, Hungary; 5R&D Laboratory, Soft Flow Ltd., Pellérdi Street 91/B, H-7634 Pécs, Hungary; 6Institute of Experimental Medicine, Laboratory of Molecular Neuroendocrinology, Szigony Street 43, H-1083 Budapest, Hungary; 7Institute of Genetics and Biotechnology, Department of Microbiology and Applied Biotechnology, Hungarian University of Agriculture and Life Sciences, Agribiotechnology and Precision Breeding for Food Security National Laboratory, Szent-Györgyi Albert Street 4, H-2100 Gödöllő, Hungary

**Keywords:** mycotoxins, analytical methods, ochratoxin A, exposure, accumulation, HPLC-FLD, ELISA, flow cytometry

## Abstract

Mycotoxins are bioaccumulative contaminants impacting animals and humans. The simultaneous detection of frequent active exposures and accumulated mycotoxin level (s) in exposed organisms would be the most ideal to enable appropriate actions. However, few methods are available for the purpose, and there is a demand for dedicated, sensitive, reliable, and practical assays. To demonstrate the issue, mice were exposed to a relevant agent Ochratoxin A (OTA), and accumulated OTA was measured by fine-tuned commercial assays. Quantitative high-performance liquid chromatography with fluorescence detection, enzyme-linked immunosorbent assay, and flow cytometry assays have been developed/modified using reagents available as commercial products when appropriate. Assays were performed on excised samples, and results were compared. Accumulated OTA could be detected and quantified; positive correlations (between applied doses of exposure and accumulated OTA levels and the results from assays) were found. Dedicated assays could be developed, which provided comparable results. The presence and accumulation of OTA following even a short exposure could be quantitatively detected. The assays performed similarly, but HPLC had the greatest sensitivity. Blood contained higher levels of OTA than liver and kidney. We demonstrate that specific but flexible and practical assays should be used for specific/local purposes, to measure the exposure itself and accumulation in blood or organs.

## 1. Introduction

Climate change, as a global problem of industrialization, induces various issues impacting wildlife, domestic animals, and human populations in contact with them [1]. This gives rise to problems and challenges due to the continuous change in formation, presence, dynamics, and distribution of natural and anthropogenic environmental contaminants. Many persistent organic pollutants (POPs) are semi-volatile, bioaccumulative, stable, toxic and might have the ability to interfere with the physiology and/or behavior of exposed organisms [2]. POPs affect immune, neural, and/or endocrine elements following a long-term or even shorter acute exposure [3,4,5]. Contaminants are now ubiquitous and can also be abundantly found in tissues of organisms [6,7].

Mycotoxins (e.g., aflatoxins or ochratoxins) are POPs and act as natural contaminants. They are secondary metabolites of specific fungi and cause severe damage inter alia to agricultural products, persisting in food and the food supply chain [8]. Exposure routes can be variable, but organisms can be affected mainly orally, consuming—accidentally or even continuously—contaminated foodstuff [9,10]. Previous studies show that increased temperature, elevated CO_2_, extremes in water availability, and other factors due to climate change influence the occurrence and/or frequency of mycotoxin production, accumulation, and contact with organisms. Consequently, this increases the risks for contamination [11]. In addition, mycotoxins are now a challenge to the security of animal health and welfare, as well as productivity, causing significant economic losses and damage.

Increasing the relevance of this topic, most mycotoxins might have toxic and carcinogenic effects on farm animals and humans [12]. Co-occurrence of mycotoxins in an exposure also can enhance the potential health risks. In addition, multiple agents that occur and act simultaneously may be capable of disturbing additively or synergistically [13]. Therefore, the detection (exact measurements) of all contaminants can be highly important. An appropriate/ideal (or “final”) solution should be able to detect the mycotoxins concurrently and be dedicated possibly to solving the specific local problem [14,15,16]. Additionally, the detection of both exposure (e.g., mycotoxin contamination in food or feed) and the levels of accumulated agents (e.g., in relevant organs or body fluids) might be most useful. It is important to find and link the source, the causes, the facts (accumulated levels), and ideally the induced biological effects. The legal limits imposed on the environmental concentrations of POPs are largely based on the results of investigations relating to toxic amounts and much less on subtoxic concentrations [2]. The duration of exposure is also crucial. In view of the ability of POPs to accumulate, some long-term effects of POPs can have serious consequences [2,8]. Even these legally neglected, low doses can reach an accumulated threshold, constituting real hazardous agents [2,6]. Thus, detection of low levels might also be important [2].

Ochratoxins are by-products of *Penicillium* and *Aspergillus* filamentous fungi of which mainly the Ochratoxin A (OTA) form exerts hazardous effects on organisms [17,18,19,20]. OTA is a widely spread mycotoxin, causing major health risks [12,21]. Generally, the mean OTA contamination levels in European food commodities are relatively low (ppb to ppm); however, elevated concentrations can occur in individual batches. Even though preventive measures are taken to keep the levels of OTA in food as low as reasonable, a certain degree of contamination seems unavoidable. Therefore, dietary exposure to OTA represents a severe health risk; OTA has been associated with human and animal diseases, including poultry ochratoxicosis, porcine nephropathy, human endemic nephropathies and urinary tract tumors [22]. OTA was shown to be carcinogenic in rodents and was classified as a Group 2B human carcinogen [23]. Besides its carcinogenic and nephrotoxic impacts, it was found to possess hepatotoxic, teratogenic, neurotoxic, genotoxic, and immunotoxic effects in animals. Extensive research has been performed to investigate its mode of action; however, this is still under debate [22,24]. Therefore, studying the kinetics of absorption, distribution, metabolism, and excretion of xenobiotics is an important tool in the extrapolation of animal toxicity data for human risk assessment [25].

The typically used analytical methods have been reviewed in recent years [20,22,25]. These methods are capable of detecting and distinguishing between the various OTA family members and metabolites but are technically and/or financially demanding. In contrast, immunoassay-based methods are typically less expensive but can usually detect one specific mycotoxin [26]. Most of these assays have been developed to detect OTA [22,27,28]. Although official monitoring programs preferentially use analytical methods such as high-performance liquid chromatography with fluorescence detection (HPLC-FLD) and liquid chromatography-tandem mass spectrometry (LC-MS/MS), immunoassay-based methods can be equally sensitive and reproducible [22,26]. To satisfy these detection limits, conventional detection methods such as enzyme-linked immunosorbent assay (ELISA), thin-layer chromatography, and HPLC-FLD have been currently employed. Some of today’s technologies are capable of easy performance of multiplex measurements of mycotoxins, including OTA, such as flow cytometry (FCM) [14,15,29]. Concomitantly, more sensitive, simple, and portable detection methods are being introduced through antibodies, aptamers, and nanomaterials [30,31,32].

Our team has been focusing on a wide examination of OTA exposure using low and high doses [33,34]. OTA has been selected as a model of POPs to emphasize the importance of such exposure and the accurate detection of both the exposure and even accumulated levels of a potentially toxic substance in the organisms.

We aimed to develop dedicated immunoassay-based solution(s) which can be compatible with standardized/most widely accepted and applied HPLC-FLD. However, the solution(s) should be practical to operate, cost-effective, as rapid as needed, accurate, reliable, with high-throughput (e.g., ELISA), and even might be a base for detection of co-occurrence of contaminants (e.g., FCM). Using such assays, we aimed to demonstrate the possibility and importance of detecting accumulated level (s) of acting mycotoxin POPs in the main organs of mice exposed orally to OTA. Since assay development tends to be a very complex and time-consuming process, at this point we focused on finding and proving the applicability and adaptability of different techniques for the aforementioned purpose. For this, we partially used commercial reagents, in order to develop assays that can be built upon. We used existing solutions for detection of exposure (e.g., contamination in food) to ease/speed up risk management procedures. To the best of our knowledge, this is the first report that describes a direct comparison of HPLC-FLD, ELISA, and FCM techniques to detect/quantify OTA accumulation using mice tissues (kidney and liver) as well as blood samples.

## 2. Results

### 2.1. Vehicle Controls and Untreated Animals

No statistical differences in any parameter were found between the untreated (absolute control) and vehicle control (VC) animals. Therefore, only data on the VC animals are given as controls below in subsections of Results.

### 2.2. Determination of Toxicity of Applied Exposures, Food, and Tap Water Consumption

The body weight of examined mice was monitored and measured during the experimental duration. Furthermore, mice were sacrificed at the end of the 3-day exposure period, and their kidney and spleen mass were recorded for further comparison (Figure 1).

There was no unprecedented modification in the body weight of the OTA-1 (exposed to 1 mg/kgbw)—and OTA-10 groups (exposed to 10 mg/kgbw), as opposed to the VC group (Figure 1a). Upon the OTA challenge (OTA-1 and OTA-10 groups), no considerable changes could be seen between the initial and final body weight of mice (Figure 1a). Furthermore, within the average mass of the spleen and kidney of OTA-exposed groups, there were no noticeable differences compared to the VC group (Figure 1b). Consequently, these observations were further strengthened by the normalized weight of examined organs (Figure 1c), where the weight of organs (Figure 1b) from VC or OTA-exposed groups were normalized to their body weights (Figure 1a). Quantified values (Figure 1a–c) are also presented in Appendix A. Consumption of food and water did not change as an effect of any applied OTA treatments. Data are therefore not shown.

### 2.3. Accumulation and Evaluation of OTA in Mice Tissues and Plasma Employing HPLC-FLD, ELISA, and FCM Methods

To offer an alternative to the HPLC-FLD method, an ELISA immunoassay, as well as an FCM-based competitive fluorescent microsphere immunoassay system, have been developed (detailed in the Materials and Methods section). For the assessment of the OTA accumulation, diverse exposure conditions (1 or 10 mg/kgbw as OTA-1 and OTA-10 groups) were used, followed by further analyses using HPLC-FLD (Figure 2(aA)) or ELISA (Figure 2(aB)) or FCM (Figure 2(aC)). For the comparison throughout the experimental interval, VC animals served as vehicle control (Figure 2a–c,A–C).

In blood plasma, the concentration of accumulated OTA was markedly elevated upon the 10 mg/kgbw OTA exposure (OTA-10 group), which was revealed by not only HPLC-FLD (Figure 2(aA), 16.123 ppb) but also the newly designed approaches (ELISA: Figure 2(aB), 10.628 ppb, and FCM: Figure 2(aC), 10.995 ppb). Furthermore, we observed significant differences between VC and 10 mg/kgbw OTA exposed animals using HPLC-FLD (** *p* < 0.01, Figure 2(aA): F_2,9_ = 4.327), ELISA (*** *p* < 0.001, Figure 2(aB): F_2,9_ = 3.099), and FCM methods (*** *p* < 0.001, Figure 2(aC): F_2,9_ = 3.092). By HPLC-FLD and ELISA, in the plasma of 1 mg/kgbw OTA treated mice, we detected apparent but barely raised OTA concentration without any significant difference (Figure 2(aA,B). Greater variation could be seen using FCM (* *p* < 0.05, Figure 2(aC)). This tendency or pattern proved that ELISA and FCM techniques are similarly suited for OTA detection (Figure 2(aA–C)).

In the kidney, we also monitored the accumulated OTA concentration using HPLC-FLD (Figure 2(bA)), ELISA (Figure 2(bB)), and FCM (Figure 2(bC)) techniques. Coinciding with the blood plasma results (Figure 2a), corresponding outcomes were also detected. We found significant differences between the kidney of VC and 10 mg/kgbw OTA exposed animals (OTA-10 group) by all methods (Figure 2(bA): *** *p* < 0.001, F_2,10_ = 8.047, Figure 2(bB): F_2,10_ = 1.978 and Figure 2(bC): F_2,10_ = 7.187). However, there was no significant difference between the kidney of VC and 1 mg/kgbw OTA exposed mice (Figure 2b). Upon the 10 mg/kgbw OTA treatment, the greatest accumulated amount was substantiated by HPLC-FLD (Figure 2(bA)), consistent with HPLC data obtained from plasma (Figure 2(aA)). This means that in kidney, the quantity of accumulated OTA was remarkably reduced (Figure 2b, app. 1.838 ppb by HPLC, 1.108 ppb by ELISA, and 1.351 ppb by FCM) in comparison to plasma (Figure 2a, 16.123 ppb by HPLC, 10.628 ppb by ELISA and 10.995 ppb by FCM).

We also monitored OTA accumulation in the liver of mice after exposure,and investigated it by corresponding techniques (Figure 2(cA–C)). The OTA accumulation pattern was overwhelmingly analogous to blood plasma (Figure 2(aA–C)) and kidney (Figure 2(bA–C)). By all methods (Figure 2(cA–C)), increased OTA occurrence could be already visible in the 1 mg/kgbw OTA exposure (Figure 2(cA), 0.291 ppb by HPLC; Figure 2(cB) 0.583 ppb by ELISA and Figure 2(cC), 0.353 ppb by FCM) as opposed to VC animals, although without any significance (except Figure 2(cB)). Significant alterations could be predominantly seen between the liver of VC and 10 mg/kgbw OTA treated mice by HPLC (Figure 2(cA); 1.676 ppb, *** *p* < 0.01, F_2,10_ = 121.5), ELISA (Figure 2(cB), 1.296 ppb, ** *p* < 0.01, F_2,9_ = 1.285) and FCM (Figure 2(cC), 1.574 ppb, **** *p* < 0.0001, F_2,10_ = 6.715).

Representative standard curves (Appendix A) and histograms (Appendix A) of one FCM measurement can be found in Appendix A. In parallel, a standard curve of one ELISA measurement (Appendix A) and representative chromatograms of HPLC-FLD (Appendix A) measurements can also be seen in SI.

### 2.4. Comparison of Methods and Correlation

To evaluate the relationship between HPLC-FLD and ELISA (Figure 3a–c,A), HPLC and FCM (Figure 3a–c,B), or FCM and ELISA (Figure 3a–c,C), corresponding samples were employed (please see Results, Section 2.3).

Using plasma samples, a particularly high correlation significance was seen between HPLC-FLD and ELISA (Figure 3(aA), r = 0.9664, R^2^ = 0.9339, CI 95% = 0.8813 to 0.9908, **** *p* < 0.0001) methods. Rather similar, very high significant correlation was also obtained from the results of HPLC-FLD and FCM (Figure 3(aB), r = 0.9428, R^2^ = 0.8888, CI 95% = 0.8037 to 0.9842, **** *p* < 0.0001). Furthermore, using plasma samples, positive and relatively strong correlation was assessed by the comparison of FCM to ELISA techniques (Figure 3(aC), r = 0.8933, R^2^ = 0.7980, CI 95% = 0.6554 to 0.9699, **** *p* < 0.0001).

From the perspective of liver tissues, acquired data of HPLC-FLD and ELISA methods revealed rather clear as well as significant resemblance (Figure 3(bA), r = 0.8513, R^2^ = 0.7247, CI 95% = 0.5423 to 0.9574, *** *p* < 0.001), underscoring their correlation. Linked to HPLC-FLD with FCM data, their tight association could be also reinforced (Figure 3(bB), r = 0.9149, R^2^ = 0.8370, CI 95% = 0.7338 to 0.9746, *** *p* < 0.001). Last, comparing FCM data to ELISA measurements, the correlation was negligibly weaker, but conspicuously discernible (Figure 3(bB), r = 0.7138, R^2^ = 0.5095, CI 95% = 0.2369 to 0.9135, ** *p* < 0.01) underlaying a positive connection.

The positive relationship between HPLC-FLD and ELISA using kidney tissues was corroborated by the correlation coefficient (Figure 3(bA), r = 0. 7341, R^2^ = 0.5390, CI 95% = 0.3076 to 0.9150, ** *p* < 0.01). The strength of the correlation indicated a very strong, positive resemblance between HPLC-FLD and FCM data (Figure 3(bB), r = 0. 9901, R^2^ = 0.9802, CI 95% = 0.9661 to 0.9971, **** *p* < 0.0001) and finally, the correlation coefficient referring to FCM and ELISA also showed a positive, high relationship (Figure 3(bC), r = 0.7001, R^2^ = 0.4902, CI 95% = 0.2428 to 0.9028, ** *p* < 0.01).

## 3. Discussion

Mycotoxins, as natural POPs, induce various challenges for our decade, affecting the health of living organisms and posing major health risks [21]. Food and feed can be frequently contaminated with moderate, or sometimes high(er), doses of such agents and the consumption of related foodstuff also transfers (as an exposure) the POPs, e.g., mycotoxins, into wildlife, domestic animals, or humans [24]. Moreover, accumulated levels in organs or tissues of herbivores can be a potential source for carnivores, omnivores, and humans. Therefore, mycotoxins can reach higher and higher levels in the food chain [35]. Due to cumulative industrial and agricultural activities, the climate has significantly changed with related abiotic factors such as average temperature and rainwater [11]. These changes affect the biodiversity of the mycotoxin-producing fungi and, therefore, mycotoxin production (period and majority), distribution, patterns, co-occurrence, and/or various types of contact exposure to mycotoxins are permanently altered [11,36]. At the extreme, new/different exposures to mycotoxin (s) or newly exposed targets can also occur [11,37].

To emphasize and model this issue, we selected and applied oral treatments on laboratory mice with different (but still realistic) doses of a frequent POP mycotoxin, OTA (Appendix A). The relevance of OTA is extremely high since it has nephrotoxic, immunotoxic, and teratogenic effects and was classified as a possible carcinogen to humans (Group 2B) [23]. The consequences of exposure to higher doses of POPs (e.g., mycotoxins as OTA) have been studied to date. However, the exposure itself is or can be generally detected (measured by an assay or commercial product). In our opinion, the prevalence of these compounds in the environment and their potential to adversely affect wildlife and human populations should receive more recognition among scientists, policymakers, and public members [38]. Consideration of the lower (legally neglectable) doses and/or measurements of low levels of POPs accumulated in exposed individuals should also be enhanced. Such doses should also be detected and measured with highly (enough) sensitive procedures.

In our treatments, we dealt with the use of both moderate and (typically accidentally occurring) higher doses (10 mg/kgbw) types of exposure to OTA (Figure 1; Appendix A). Most of the existing commercialized solutions mainly characterize exposure (e.g., contamination of foodstuff, cereal grain, or maize), and few procedures exist for the detection of accumulated mycotoxins or both [39,40]. The TOXI-WATCH Kit is validated for corn and wheat, as is the MycoFoss^TM^ solution. Some commercially available kits can deal with serum or supernatant from an in vitro culture. Procedures can be expensive, long, and/or require specific knowledge, laboratory background, or sophisticated instrumentations such as HPLC-FLD [41]. There is, therefore, increasing demand for quick, easy, cheap, effective, sturdy, and safe (QuEChERS) procedure (s) [42]. In this recent study, we wanted to demonstrate that appropriate, QuEChERS assays dealing with blood, organs, and tissues (and ideally with the exposure) could be rapidly developed and applied for our purposes (Figure 2 and Figure 3). In addition, we wanted to include assay on more platforms in case of use for future objectives. Instrumental analyses generally boast good accuracy and reproducibility, including HPLC-FLD, LC-MS/MS, and ELISA, which have been employed to accomplish the analyses within a relatively short time [42,43].

Nowadays, among the available conventional methods (e.g., HPLC) the immunochemical ELISA has been traditionally applied [41,44,45]. Qualitative detection methods include membrane-based immunoassays, lateral flow systems, and immunoaffinity columns. Each is commercially available and may be used to screen samples rapidly. Reading is (usually) visual, although the results may be quantitative when using appropriate devices, such as a scanner [27].

Similarly to other mycotoxins, the structure of OTA is very stable even at high temperatures and resistant to hydrolysis; hence the processing of raw materials in the feed and food industry does not eliminate the OTA, and the toxin can remain intact in the end-products [33]. Therefore, the OTA is often found as a contaminant in cereal grains or other crops and plant products such as red wine, coffee beans, peanuts, cocoa beans, and different spices [33]. Although traditional chromatographic and immunoassays appear mature enough to attain sensitivity up to the regulation levels, alternative detection schemes are still being enthusiastically pursued to meet the requirements of rapid and cost-effective detection [42] as well as to detect legally acceptable levels or their accumulation in targets.

OTA exposure determines various toxicological effects, including the disruption of gut microbiota homeostasis, hepatotoxicity, genotoxicity, immunotoxicity, embryotoxicity, neurotoxicity, testicular toxicity, and nephrotoxicity [20]. Most studies have been focused on the immunotoxicity of OTA, such as inflammation, and on the kidney and liver as the organs which are mainly affected by the exposure [20,44,46,47,48,49]. Earlier, we have also found that the spleen and kidney can be sensitive indicators of OTA toxicity, and some related mechanisms have also been detected [33,34]. Currently, we have not focused on further understanding of any induced effects and biological mechanisms. However, the recognition of symptoms following a toxic OTA exposure was important for us. Our study included general toxicological routines performed to characterize the 3-day exposure to OTA. Its toxicity can commonly be monitored via changes in body or organ weight, the activity of liver-specific enzymes (not measured here), or other Impact-specific symptoms [50,51]. However, the body and organ (Figure 1b, Appendix A) weights did not reveal significant differences between the OTA-exposed groups and the VC group.

The same phenomenon was found when we presented normalized organ weight vs. OTA exposure (Figure 1c, Appendix A). More precisely, we failed to demonstrate additional signs of toxic OTA contamination as described in the literature [20,21]; however, a similar result was noticed in a different study by our work team [33]. We also have to mention that we used a short (3-day) exposure period compared to most in vivo studies in the literature [20]. Otherwise, these findings are likely to infer that the doses and applied duration of OTA exposure could still be considered subclinical. Rather similar outcomes were obtained in the acute OTA toxicity rat model, where no correlation was observed between OTA exposure and nephrotoxicity or hepatotoxicity [52]. In a study, kidneys and muscle samples from exposed pigs were tested for the presence of OTA [24,44]. Both competitive indirect ELISA and HPLC-FLD were used to determine OTA levels. The mean and median values for OTA in kidneys were 0.29 and 0.25 ppb, respectively. Mean and median values found for muscle (0.024 and 0.01 ppb) are significantly lower than those reported in other studies demonstrating that, irrespective of the geographical provenance of pigs, OTA incidence is far from representing a real concern for consumers. Results obtained plotting ELISA vs. HPLC-FLD show that ELISA tends to slightly underestimate the OTA content compared to HPLC; nevertheless, ELISA remains a proper tool as a rapid screening (semi)quantitative technique [44]. Elsewhere, a good agreement was found between OTA levels in muscle and liver measured by HPLC (liver concentration = 2.9 × OTA muscle concentration, r = 0.981) obtained from French pigs reared following three different types of production (organic, Label Rouge, and conventional) [53].

Some of the excised organs (liver and kidney) and blood as a basic compartment for transfer of consumed/adsorbed elements in an organism have also been included in our experiments for OTA contamination/accumulation, as described in Section 2.3. Therefore, depending on the exposure to OTA, its presence and accumulation occurred and could be quantitatively detected with all developed assays, namely HPLC-FLD, ELISA, and FCM (Figure 2).

After oral administration of various doses of OTA in experiments, significant elevation of the OTA concentrations was found in plasma (Figure 2a). OTA levels above the detection limit were found even in control animals, which could be attributed to unavoidable natural contamination, as reported previously [34]. Acute (sub-chronic, 3-days) exposure to 10 mg/kgbw OTA resulted in significant levels in both kidney and liver compared to control (VC). Exposure to a lower level of OTA (1 mg/kgbw) could also lead to a slightly increased level; however, that accumulation was not statistically significant. ELISA and FCM generally showed lower levels than HPLC-FLD could measure (in both liver and kidney), although recoveries on applied OTA spiking were appropriate (Figure 2b,c). For this purpose, quantitative assays on three different platforms have been developed (Appendix A). Assays operate with respective OTA extraction of the sample (resulting in different efficiencies), but internal spiking was continuously applied, resulting in acceptable recoveries. ELISA and FCM measurements on blood correlate properly to HPLC-FLD and to each other (Figure 3a, Appendix A). In addition, strongly positive correlations between HPLC and ELISA, HPLC and FCM, and ELISA and FCM were found in kidney and liver tissue measurements (Figure 3b,c, Appendix A). The observed slight differences in OTA concentration levels measured by HPLC-FLD, ELISA, or FCM might originate from different reasons. The immunoassay procedures, such as ELISA and FCM, operated with (partially) different immunoreagents (e.g., biotinylated or unconjugated anti-OTA antibodies), surface/area for the reaction (on magnetic particles or a well of a polystyrene 96-well microtiter plate sensitized previously with streptavidin), diluted labeled marker (fluorescent or HRP-mycotoxin conjugates), and generally the micro-environment of the assay (buffers, diluents, standards, pH conditions, incubation time, etc.). The performance of immunoassays depends greatly on the immunoreagents (here, anti-OTA antibodies) and/or other components, which is a basic difference in comparison with the HPLC-FLD method, where the use of binding agents is not necessary (it might be required for target concentrating or target removal from matrix elements).

Assays are paired with various detections (absorbance reader or flow cytometer) equipped with different working essence and sensitivity. A cytometer depends on its optical features, can detect 25–100 molecules of equivalent soluble fluorochrome on a bead/particle with the appropriate size; allowing even the application of highly diluted conjugates which might be needed for a sensitive competitive assay [14,15]. Measurements of thousands of particles in an assay as “individual replicates” increase the reliability of the procedure and decrease the dependency on general technical issues such as pipetting/manual mistakes, improper washing, etc. The ELISA usually can be less sensitive in comparison with FCM and has a higher dependency on the operator or assay environment. However, FCM and ELISA are more “practical”, generally faster measurements with much easier and cheaper procedures in comparison with HPLC-FLD, where the high-throughputness (mainly with ELISA) or capability for multiplex measurements (2–50 targets simultaneously, mainly with FCM) could be easily achieved in the future. We usually measured the lowest concentrations with FCM (Figure 2). ELISA measurements also provided lower amounts of OTA than HPLC-FLD, but concentrations were usually higher than FCM (Figure 2). Dynamisms of assays and recognition of targets by the binding agent differ between procedures. The applied FCM method (as the main advantage) dealt with very short incubation (minutes). Further fine-tuning of assay or pretreatment is needed to have (more) proper recoveries of the target compound.

Additionally, applied extraction procedures and other pretreatments for reducing the sample matrix effect prior to assay have also been different in HPLC-FLD, ELISA, and FCM. Therefore, further optimization/fine-tuning and standard validation on developed assay might be still required to help assay performance be improved. For example, OTA might be more efficiently extracted from the matrix using harsh/organic solvents such as ACN, chloroform, or methanol. However, such chemicals are usually hazardous and can negatively influence the functionality of binding agents and/or conjugates, and thus even assay performance. Finding a balance between the most efficient extraction and optimal assay performance can be a key for assay development.

We also aimed to demonstrate that assay(s) to meet local needs and measuring techniques even for low doses can be rapidly developed. Ideally, the assay development time could be reduced with a further fine-tuning of an existing and commercially available solution; a modification of an ELISA Kit/product (e.g., TOXI-WATCH), or even an assay operating in an automated, instrumentally supported, dedicated procedure (MycoFoss™). Those solutions were originally dedicated to characterizing exposure (mycotoxin level in contaminated foodstuff); now, they can also measure accumulated OTA levels in exposed organisms. We developed ELISA and FCM-based assays because future applications might differ depending on the necessity for high-throughputness and/or multiplex measurement to detect the co-occurrence of mycotoxins or other POPs. Although some ELISA-like solutions, such as low density protein/antibody microarrays, can also be applied as an assay, flow cytometry looks to be an ideal platform for multiplex measurement. However, the complexity of a flow cytometer and the need for a trained operator are disadvantageous; therefore, process automation of an FCM-like solution might be a proper way for immunoassay measurement of mycotoxins or any other POPs whether as exposure or as accumulation in the exposed organisms.

## 4. Conclusions

We conclude that dedicated assays on different technical platforms (HPLC-FLD, ELISA, FCM) have been developed for measuring OTA in blood and the main organs of orally exposed laboratory mice. Assays were applied on obtained samples and provided comparable results. The presence and accumulation of OTA in blood, liver, and kidney following a 3-days exposure could be quantitatively detected by all developed assays. Assays performed similarly, but overwhelmingly HPLC gave reliable measurements at the lowest exposure to OTA. Additionally, blood contained the highest level of OTA; mycotoxin accumulation in the organs examined was lower, as expected. We demonstrate that specific but flexible, practical assays are needed and can be built for actual/local purposes to measure exposure levels and uptake by exposed organisms.

## 5. Materials and Methods

### 5.1. Chemicals and Materials

Chemicals and materials were obtained from Sigma-Aldrich (St. Louis, MO, USA), the Invitrogen Corporation (Carlsbad, CA, USA), or the BD-Biosciences (San Jose, CA, USA) unless otherwise indicated. All chemicals used were of analytical grade. OTA mycotoxin (CAS registry number: 303-47-9) applied for exposure in crystalline powder form was purchased from Fermentek Ltd. (Jerusalem, Israel) and kept refrigerated at −20 °C until use.

### 5.2. Animals

Medically certified CD1 male mice (Institute of Experimental Medicine, Budapest, Hungary) weighing 35–40 g, aged 7–9 weeks, originating from different litters, were used at the beginning of the experiments. After the arrival, the animals were housed individually in standard plastic cages, assigned to various experimental groups (as described below), and kept under controlled conditions of temperature at 21 ± 1 °C and relative air humidity at 55–65%. The animals kept under an automated, 12-h dark-light period (lights on at 7:00 a.m.) in the conventional/minimal diseases animal house of the Institute. Laboratory rodent chow (CRLT/N, Charles River, Gödöllő, Hungary) and the tap water were ad libitum available. This diet contained 86% dry matter, 19% crude protein, 17% digestible protein, 4.5% crude fat, 6% crude fiber, 6% crude ash, 40% nitrogen-free extract, 0.8% calcium, 11,000 IU/kg vitamin A, 600 IU/kg vitamin D3, and other amino acids lysine, methionine, and cysteine. The used tap water was chemically analyzed by the Water Company of the Local Government of Budapest City. Batches of rodent feedstuff were tested for OTA by the Laboratory of Soft Flow Hungary R& D Ltd. (Pécs, Hungary), using their commercial ELISA-based kit (TOXI-WATCH Ochratoxin A ELISA Kit, Cat. Nr.: 3000051) and traces of OTA were detected (2.1 ± 0.11 ppb).

The acclimatization of animals lasted for 2 weeks. After that, the care and the procedures were carried out in accordance with the European Communities Council Directive of 11/24/86 (86/609/EEC), and the experimental protocol (mentioned below) was approved by the Institutional Animal Care Use and Committee of the Institute of Experimental Medicine, Budapest Hungary (PEI/001/35-4/2013).

### 5.3. Exposures, Experimental Groups, and Protocol

The testing parameters and the dosage and duration of the exposure to OTA were selected according to Luhe et al. [13] and related legislation.

As an expositor agent, OTA in 1 mL of% aqueous dimethyl-sulfoxide (DMSO) finally diluted in sterilized tap water was administered daily by gavage in a final dose of 0, 1, or 10 mg/kgbw. Such doses are lower than oral LD50 in rats (28 mg/kgbw [53]) and mice (53 mg/kgbw [54]). For exposure preparation, the OTA was first dissolved in DMSO, and a master mix in 100 mg/mL OTA content was diluted and stored at 4 °C. That master mix was then subsequently diluted with sterile tap water (containing 10 mM Tris, pH = 8.0) to reach the selected OTA dose. The mice were exposed to OTA for 3 days (1 mg/kgbw, *n* = 5, hereinafter OTA-1 group, or 10 mg/kgbw, *n* = 4, hereinafter OTA-10 group) or to vehicle only (as vehicle control) for 3 days likewise (*n* = 5, hereinafter VC group). Absolute controls were unexposed (*n* = 5, hereinafter AC group). Individual body weight, daily food, and tap water consumption were measured individually. Exposure was applied once daily (200 µL/animal) in the morning. After the last exposure of an animal, the subject was immediately killed by rapid decapitation, and its trunk blood was collected into Vacutainer^®^ EDTA tubes for further measurements. Plasma was separated accordingly and kept refrigerated at −70 °C until measurements, as described below. Additionally, main organs such as the whole spleen, kidneys, and most of the liver, muscle, or fat were carefully removed, and the wet weights of organs were measured. Laboratory samples were kept at −70 °C until starting measurements, as described below.

Based on our prior studies, to monitor morphological alterations standard examinations were performed on animals or on their excised organs throughout and/or at the endpoint of the experiment (data not shown) [33,34].

### 5.4. Measurements Using HPLC-FLD Method

Related experiments, setting of protocol, performing extraction of OTA from samples, and final measurements on OTA levels and its accumulation in organs were performed by the workmates of the Department of Pharmacology, Faculty of Pharmacy, University of Pécs (Pécs, Hungary). HPLC-FLD methodology was accepted as standard [55] and used as a basis for comparison.

#### 5.4.1. Extraction of OTA Applied

A two-fold volume of acetonitrile (ACN) was added to mice plasma. Then samples were vortexed, sonicated (3 min), and centrifuged (5 min, 14.000× *g*, 4 °C). The supernatant was diluted two-fold with the HPLC eluent (sodium borate buffer (10 mM, pH 10.0) and ACN, 87:13 *v*/*v*%) and then directly analyzed. After weighing kidney and liver samples, a two-fold amount of water was added, then tissue samples were homogenized using a Potter–Elvehjem glass-teflon homogenizer. After that, a 200 μL volume of ACN was added to 100 μL homogenate; then, tissue samples were treated the same way (vortexing → sonication → centrifugation → two-fold dilution with the HPLC eluent) as we described above in regard to the plasma samples.

#### 5.4.2. Measurements

An integrated HPLC system (Jasco, Tokyo, Japan) was applied, which is built up from an autosampler (AS-4050), a binary pump (PU-4180), and a fluorescence detector (FP-920). Chromatographic data were evaluated employing ChromNAV2 software (Jasco, Tokyo, Japan). OTA was quantified as it has been recently reported [55]. Briefly, samples (20 μL) were driven through a SecurityGuard precolumn (C18, 4.0 × 3.0 mm; Phenomenex, Torrance, CA, USA) linked to a Kinetex EVO-C18 (150 × 4.6 mm, 5 μm; Phenomenex) analytical column with 1.0 mL/min flow rate, at room temperature. Isocratic elution was performed, where sodium borate buffer (10 mM, pH 10.0) and ACN (87:13 *v*/*v*%) were applied in the mobile phase. OTA was detected at 446 nm (λ_ex_ = 383 nm). Samples were measured in triplicates.

The linearity of the method was determined between 10 nmol/L and 1.0 μmol/L (4.0–403.8 μg/L) concentrations (R^2^ = 0.9996). Limit of detection (2 nmol/L or 0.8 μg/L) and limit of quantification (6 nmol/L or 2.4 μg/L) values were defined as the lowest concentrations when the signal-to-noise ratios reached 3 and 10, respectively. The intraday repeatability was evaluated based on the intraday coefficient of variation (0.55%; *n* = 7).

#### 5.4.3. Data Generation and Interpretation of Results

The recovery in plasma and tissue samples was 90.6 ± 2.4%. OTA concentrations were calculated from the area under the curve (AUC) values using a calibration line (10 nmol/L to 1.0 μmol/L), after which data were corrected with the previously determined recovery.

### 5.5. Measurements Using ELISA Method

To provide a potential alternative to the HPLC method (because of its high complexity, low throughputs, and low cost/test efficiency), an ELISA-based immunoassay procedure has been developed and applied to samples. Related experiments, setting of protocol, performing extraction of OTA from samples, and measurements on OTA levels in plasma and its accumulation in organs were done by the workmates of the Department of Animal Biotechnology, Hungarian University of Agriculture and Life Sciences (Gödöllő, Hungary).

#### 5.5.1. Extraction of OTA, Sample Preparation

Stored plasma samples were melted and diluted/extracted with three-fold of ACN/water solution (84/16, *v/v*) and shaken on an orbital shaker for 15 min at RT. Extracts were centrifuged (RT, 5 min, 8.000× *g*), and supernatants were collected and subsequently diluted with 0.01M PBS, pH = 7.4. The dilution factor on supernatant was 250×.

Usually, 0.25 g of excised organs/samples was homogenized in ice-cold, 0.5 mL of 50 mM sodium acetate buffer (pH = 4.80). First, homogenization was performed with a bench-top bead beating lysis system (FastPrep-24 Classic, MP Biomedicals, Irvine, CA, USA) using metal beads followed by incubation for 3 h at 37 °C on a shaker with the addition of β-glucuronidase/arylsulfatase from *Helix pomatia*, accordingly to the instructions of the manufacturer. Following the incubation, the homogenized sample was diluted with ACN/water solution (84/16, *v/v*) and shaken (extracted) for 15 min. Next, extracts were centrifuged (10.000× *g*, 10 min, at 4 °C), and the supernatant was separated and subsequently diluted 100× with 0.01 M phosphate-buffered saline (PBS, pH = 7.40). Finally, the diluted extracts were used for measurements with the assay.

#### 5.5.2. Measurements with ELISA-Based Immunoassay

The previously developed reagents of an existing ELISA kit/product (TOXI-WATCH Ochratoxin A ELISA Kit, Cat. Nr.: 3000051, Soft Flow, Pécs, Hungary) have been used for fine-tuning/further development of a quantitative assay working with the mouse samples. That product was originally developed and used for measurement of OTA contamination in food and feed (maize and wheat, the applied range of the measurement was 0.125–4 ppb). The existing ELISA Kit uses a proper mouse monoclonal antibody (OTA-1542 clone). Antibody was raised against OTA-bovine serum albumin conjugate and had a low cross-reactivity with Ochratoxin B (9.3%). Therefore, the same clone/antibody is used in the FCM-based solution (described below).

Standard polystyrene 96-well microtiter plate surface sensitized with streptavidin, biotinylated mouse monoclonal anti-OTA antibody, and an in-house made OTA-horseradish peroxidase (HRP) conjugate as tracer were used in a 1-h competitive assay, where OTA content of samples can compete with the tracer in a microplate-based well/compartment The assay steps were briefly the following: 50 μL of diluted, prepared extract, 50 μL of tracer and 50 μL of biotinylated antibody were added into the well/compartment. Following a 1-h incubation at 37 °C on a plate-shaker, wells were washed (280 µL/well) with the usual wash buffer (0.05% Tween-20 in 0.01 M PBS, pH = 7.40) and 150 μL 3,3′,5,5′-Tetramethylbenzidine (TMB) substrate was added. Then, 10 min later, the forming color reaction was stopped with 1N sulfuric acid (50 µL/well). Absorbance/optical density was measured in a conventional plate reader at 450 nm (reference wavelength was 630 nm), and data processing was applied. Samples were measured using triplicates. The recovery was 65.4–77.6%, measured with applied OTA spiked samples.

#### 5.5.3. Data Generation and Interpretation of Results

Raw data have been processed with acquisition software of microplate reader Biotek EL800 equipped with Gen5 software (Biotek, Santa Clara, CA, USA). Curve fitting/analysis and calculation of OTA concentrations were also done with Gen5. Spiking of untouched/prepared samples with known (low and high levels) amounts of OTA (accordingly to legislation likewise), calculating those recoveries as a general development tool, and re-validation of measurements have been continuously used.

### 5.6. Measurements Using Flow Cytometry

To provide a potential alternative of HPLC, which might be able to act as a proper basis for further easily, simultaneous measurements of OTA and another co-occurring mycotoxin, an existing, quantitative, flow cytometry-based, direct competitive, fluorescent microsphere immunoassay system/procedure (PMID: 22841575) has been further developed/modified. Related experiments, setting of protocol, performing extraction of OTA from samples, and measurements on OTA levels and its accumulation in organs were performed by the workmates of R&D Laboratory of Soft Flow Ltd. (Pécs, Hungary) [14,15].

#### 5.6.1. Extraction of OTA Applied

Extraction of samples was based on the procedure applied with the HPLC method, as described above. In addition, extracts of organs or prepared plasma samples have been diluted using their propriety standard diluent. Prepared plasma samples were diluted by 300×, and organ extracts by 25×.

#### 5.6.2. Measurements

Previously developed reagents of an existing, related flow cytometry-based commercial product (MycoFoss^TM^, Foss Analytical, Hillerod, Denmark) have been used to fine-tune a quantitative assay working with mouse samples.

This instrument was originally developed and used to measure OTA contamination in food and feed (corn and wheat, the applied range of the measurement was 5–100 ppb, in multiplex measurement). Microparticles sensitized with a mouse monoclonal anti-OTA antibody, and an in-house made fluorescent OTA-conjugate as tracer were used in a rapid, 2 min competitive assay, where OTA content of samples can compete with the tracer in a 96-well microplate-based, filter surface (1.2 μm pore size) well/compartment (AcroPrep Advance, 350 μL/well, Pall, New York, NY, USA). The assay steps are briefly the following: Wells/compartments were pre-wetted with 50 µL of wash buffer (0.01 M PBS/0.01% Tween 20), followed by aspiration with a vacuum pump. Then, 50 μL of the prepared sample was pipetted into wells together with 30 μL of finally diluted OTA tracer. Next, 30 μL of microbead conjugate was also added to the compartment, and the plate/well was shaken for exactly 2 min at 600 rpm, 40 °C. The shaking/incubation reagent mix was removed by vacuum, and unbound/unreacted components were washed out (280 µL/well). Next, beads were resuspended in PBS and subjected to flow cytometric measurement. Same-similar amounts of bead events (400–450) were collected and then analyzed as described below. Measurements were performed on a Thermo Attune NxT flow cytometer equipped with blue (488 nm), green (532 nm), and red (637 nm) lasers and plate holder/robotics. Runs used the following instrumental settings; forward scatter (FSC): 260 V, side scatter (SSC): 250 V, blue laser channel (BL1): 300 V, BL2: 525 V, BL3: 200 V, red laser channel (RL1): 220 V, RL2: 220 V, RL3: 280 V, green laser channel (GL1): 486 V, GL2: 550 V, GL3: 550 V, GL4: 550 V, Flow Options: 100 μL/min, Acquisition vol: 50 ul, Total sample vol: 180 μL, Stop Options: 50 μL, Mixing cycles: 3, Rinse options: 3. Samples were measured using triplicates. The recovery was 65.4 and 77.6%, as was measured with applied OTA spiked samples.

#### 5.6.3. Data Generation and Interpretation of Results

Data files were first processed with a dedicated post-acquisition software aimed at bead-based assays (FCAP Array Infinite, Soft Flow Ltd., Pécs, Hungary). The software is designed to detect inter alia mycotoxins included in that specific assay up to a maximum of six. If, for example, an assay is a four-plexed run, the software will automatically deal with it accordingly. There is also an automatic selection of the appropriate dynamic range for standard curve generation. The post-acquisition software also performs a curve-fitting algorithm to report each standard curve using a format that adjusts for instrument-dependent variance found in the list-mode files. Calculated MFI values relating to specific bead cluster (beads sensitized with an anti-OTA antibody with a specific color intensity) and the color of bound OTA tracer on that cluster was further processed, including curve fitting/analysis and calculation of OTA concentrations. Spiking of untouched/prepared samples with known (low and high levels) amounts of OTA (accordingly to legislation likewise), calculating those recoveries as a general development tool, and re-validation of measurements have been continuously used.

### 5.7. Statistical Analysis

Data analyses were performed by Prism v9.3.1. (GraphPad Software, San Diego, CA, USA). Parameters of weights including body, kidney, and spleen were evaluated by a mixed-effects model using Tukey’s multiple comparisons tests. Experimental data were processed by one-way analysis of the variance. Groups were compared using Fisher’s LSD post hoc test, and simultaneously the OTA groups were compared to the unexposed absolute control or VC groups using the Dunnett post hoc test. The results are presented as means ± standard error of mean (SEM) and * *p* < 0.05, ** *p* < 0.01, *** *p* < 0.001 or **** *p* < 0.0001 values were considered as statistically significant. Simple linear regression and Pearson correlation analyses were conducted with 95% confidence intervals to compare outcomes of assorted methods. Correlation coefficients (r) offered information about the strength and direction of an association between two continuous variables (r = ± 1 represents perfect, linear, and monotone relationship; r = 0: no linear or monotone relationship; r < 0: negative, or reverse relationship and r > 0: positive relationship). Significance was considered as * *p* < 0.05. Four parameter logistic standard curves were created for ELISA by Prism v9.3.1.and for FCM measurements, where R^2^ represents the goodness of fit.

## Figures and Tables

**Figure 1 toxins-14-00634-f001:**
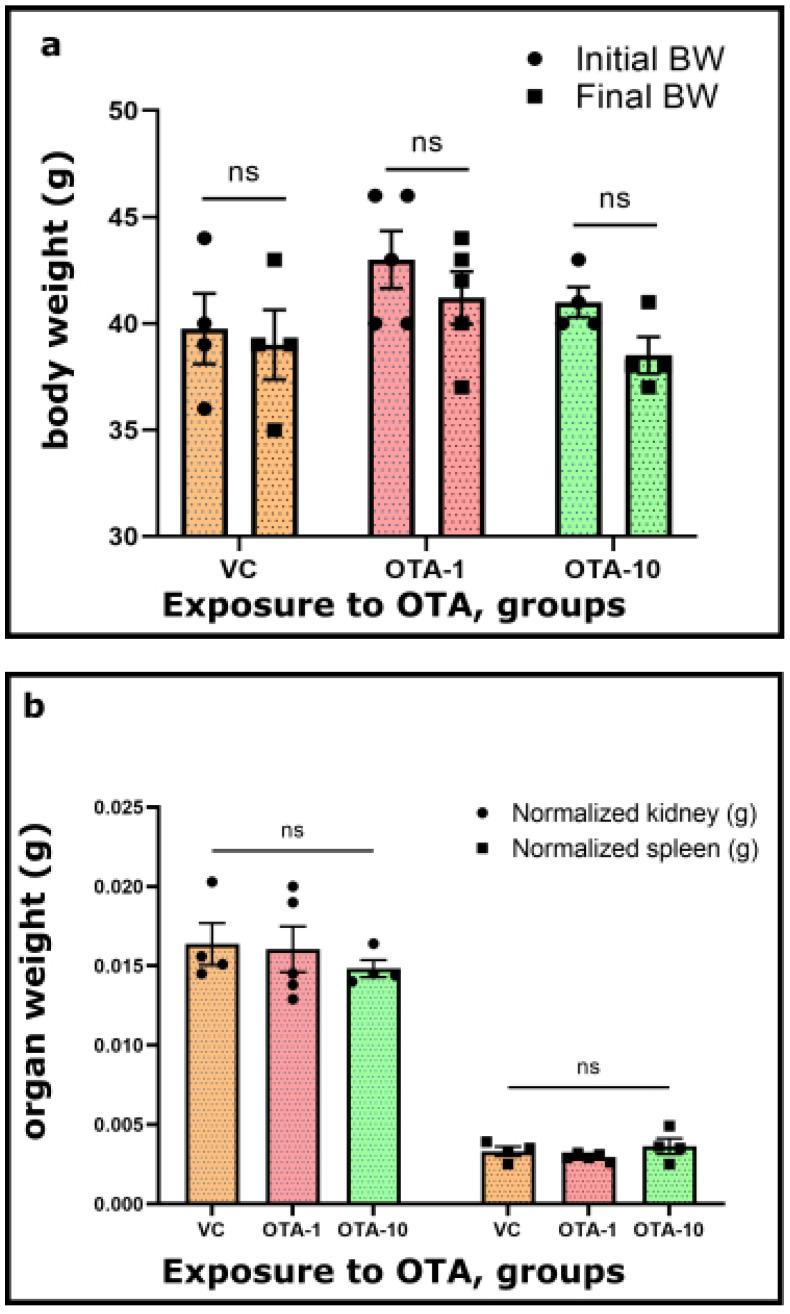
Alterations in (**a**) body weight, (**b**) mass of kidney and spleen, and (**c**) normalized weight (g/100 g body weight) of kidney and spleen from VC (vehicle control, *n* = 4) and OTA-1 (1 mg/kgbw, *n* = 5, OTA-1 group) or OTA-10 (10 mg/kgbw, *n* = 4, OTA-10 group) treated animals. A three day exposure period was chosen, and vehicle controls were also kept for the same period. Results are illustrated as means ± standard error of the mean (SEM). Signs on columns (dots and squares) denote the unique data. For the significance of the data, a mixed-effect analysis with Tukey’s post-hoc test of GraphPad Prism Software v9.3.1. was performed (*p* < 0.05). ns: not significant.

**Figure 2 toxins-14-00634-f002:**
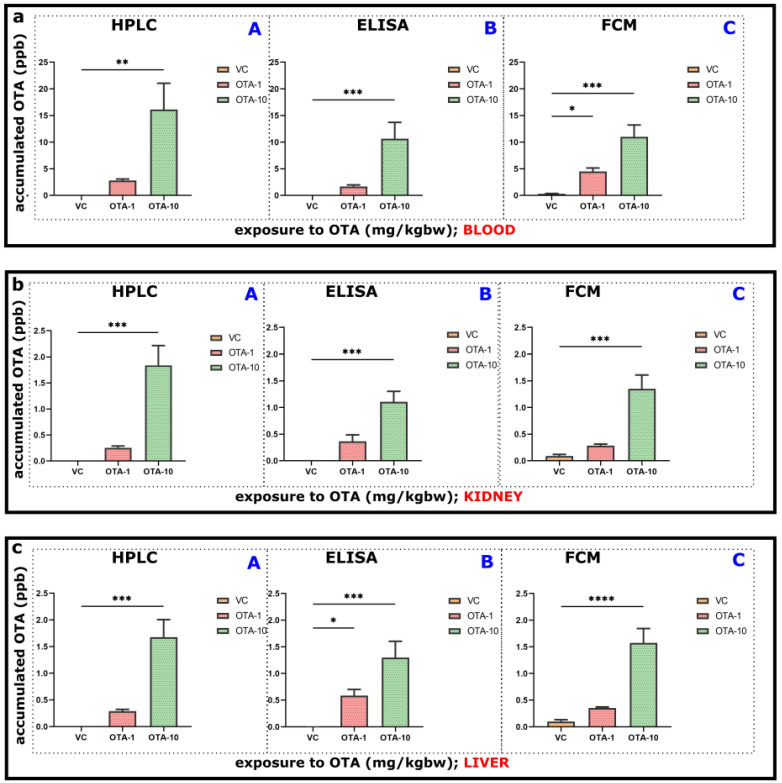
Estimation of accumulated OTA in diverse organs and blood plasma of mice. Mice were exposed to OTA for 3 days (1 mg/kgbw, *n* = 5, OTA-1 group, or 10 mg/kgbw, *n* = 4, OTA-10 groups), or to vehicle only (as VC) was used (*n* = 4, VC group). (**a**) Blood plasma (**b**) kidney and (**c**) liver of mice were scrutinized for the exact evaluation of accumulated OTA (ppb) using HPLC-FLD (**A**), ELISA (**B**), and FCM (**C**) methods. OTA-1 and OTA-10 groups were compared to the VC groups. Significance of the data was established by one-way ANOVA and OTA exposures were compared to the VC groups using Dunnett’s post-hoc test of GraphPad Prism Software v9.3.1. (* *p* < 0.05, ** *p* < 0.01, *** *p* < 0.001, **** *p* < 0.0001). The results are presented as means ± standard error of the mean (SEM).

**Figure 3 toxins-14-00634-f003:**
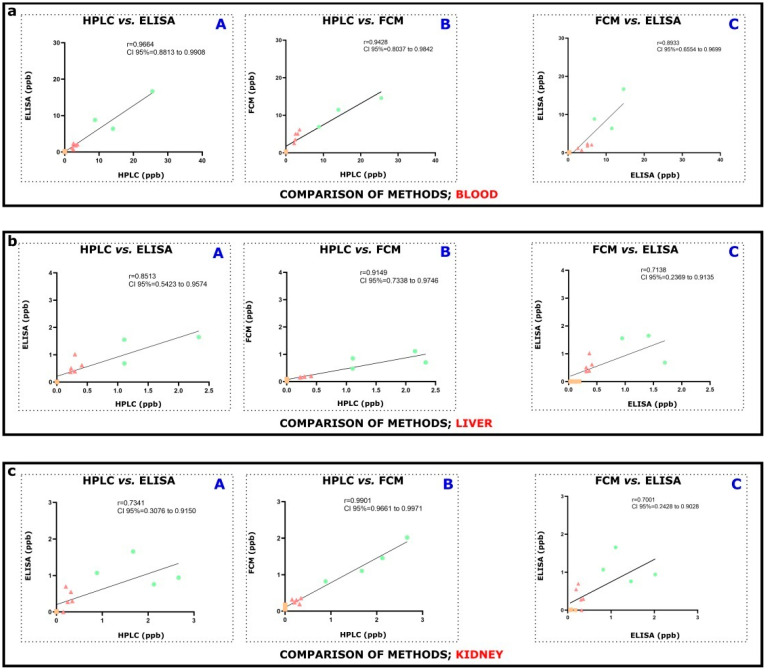
Consideration and evaluation of similarities or dissimilarities between two methodological approaches. Comparison and correlation were explored between (**A**) HPLC vs. ELISA, (**B**) HPLC vs. FCM, and (**C**) FCM vs. ELISA methodological approaches in (**a**) blood plasma, (**b**) liver, and (**c**) kidney of OTA exposed (1 or 10 mg/kgbw as OTA-1 and OTA-10 groups) or unexposed (VC) mice. In the figure, orange squares represent data of VC animals; pale red triangles show data of animals exposed to 1 mg/kgbw OTA, and green circles correspond to data of 10 mg/kgbw OTA exposed animals. Values of the x and y axes are given in ppb. Statistical analysis for (**a**–**c**) was performed using Pearson’s correlation. The correlation coefficient, r, is also indicated. Analysis and figures were made by GraphPad Prism version 9.3.1.

## Data Availability

The data presented in this study are available in the article or on request from the corresponding author.

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
