# Peer review of "Analysis and Comparison of Rapid Methods for the Determination of Ochratoxin a Levels in Organs and Body Fluids Obtained from Exposed Mice"

_toxins, 2022, doi:10.3390/toxins14090634_

Round 1

Reviewer 1 Report

In this work, quantitative HPLC, ELISA, and flow cytometry assays have been developed/modified for OTA determination in CD1 mice. This research revealed that the presence and accumulation of OTA following even a short or discrete exposure could be quantitatively detected.

This work aims to investigate OTA levels in organs and body fluids from exposed mice. There were three different analytical methods developed for OTA determination. Please explain why the three methods were compared in this work? what is the difference among the three methods? Only ELISA is a rapid method, while the other two are normal instrument methods. I would suggest the authors use a most suitable method for OTA determination in this work.

The authors mentioned in section 5 “On the other side, basic toxicological and morphological examinations were performed on animals or their excised organs throughout and/or at the endpoint of the experiment”, however, there are no related morphological results in the results.

Figure 1. It seems no significance results were observed in the figure.

Figure 2. OTA was detected in Vc group by FCM, please explain. Is it from the feed or others?

Figure 3. For methods comparison, it seems correlation coefficient (r value) is low (less than 0.8). The authors should specify the most suitable method with high reliability.

Since this work is a methods comparison, the authors should explain more about the method validation in the results section.

Author Response

Referee report from Reviewer 1

General comments:

In this work, quantitative HPLC, ELISA, and flow cytometry assays have been developed/modified for OTA determination in CD1 mice. This research revealed that the presence and accumulation of OTA following even a short or discrete exposure could be quantitatively detected.

This work aims to investigate OTA levels in organs and body fluids from exposed mice. There were three different analytical methods developed for OTA determination. Please explain why the three methods were compared in this work? what is the difference among the three methods? Only ELISA is a rapid method, while the other two are normal instrument methods. I would suggest the authors use a most suitable method for OTA determination in this work.

The authors mentioned in section 5 “On the other side, basic toxicological and morphological examinations were performed on animals or their excised organs throughout and/or at the endpoint of the experiment”, however, there are no related morphological results in the results.

Figure 1. It seems no significance results were observed in the figure.

Figure 2. OTA was detected in Vc group by FCM, please explain. Is it from the feed or others?

Figure 3. For methods comparison, it seems correlation coefficient (r value) is low (less than 0.8). The authors should specify the most suitable method with high reliability.

Since this work is a methods comparison, the authors should explain more about the method validation in the results section.

Major comments:

  1. Please, explain why the three methods were compared in this work? what is the difference among the three methods?

Author response: HPLC is a widely accepted and standardly applied analytical method for similar purposes. It is also mentioned/referred as “gold” standard application inter alia in related legislation for measurements of (for instance) mycotoxins in food and/ feed.

However, HPLC requires complex instrumentation, complicated extraction and/or measurement procedures, harsh solvents, well-trained operators and generally a laboratory background. Throughput provided is low and it is not a cost-effective process.

FCM and ELISA are immunoassay-based methods (using biochemical reagents (e.g., antibodies) mimicking the main functions of immune system of biological organisms). FCM also operates with complicated instruments (but those are now usually cheaper and easier to handle in comparison with the HPLC), but ELISA needs only a simple, cheap absorbance reader. They are high-throughput procedures, total assay time excluding (or even including) extraction can be minutes. Price per test is much lower.

FCM and ELISA methodology/measurements could be easily set, installed, and applied (even in fields!) as close to occurring necessities as possible, and can be such flexible as required. ELISA can have highest throughput, but FCM can capable easily to measure more analytes simultaneously (using bead-based technology). We wanted to demonstrate those assays can be set which provides results in a more appropriate/convenient way then HPLC.

Secondly, because of advantages of ELISA and FCM, our work team deals with developments of such assays (and needed assay components, reagents) for a long while.

In our opinion FCM or ELISA can be most proper alternative solutions to HPLC.

  1. Only ELISA is a rapid method, while the other two are normal instrument methods. I would suggest the authors use a most suitable method for OTA determination in this work.

Author response: We agree with opinion of reviewer, HPLC and FCM are “instrument methods”. We also think that currently - because of simplicity and cost-effectiveness - ELISA can be the most suitable (in this work).

However, depending on assay/quality of assay components, both ELISA and FCM can be fast, accurate and reliable. ELISA (including commercially available solutions/products) operates with low incubation time (from minutes to 1-2 hours generally). The microbead-based flow cytometry can be also correspondingly rapid technique (needs few minutes only, for instance the MycoFossTM instrument provides results within 8 minutes with extraction!). Microbeads in an assay act as “parallels” thus measuring color reaction from surface of more beads can increase reliability of assay. Emission of even low level of mycotoxin conjugates applied is detected by very sensitive instrument (flow cytometer). Nowadays, more easier or cheaper benchtop cytometers or FCM-based solutions (e.g. MycoFossTM) appear, therefore FCM might become also “most suitable” in close future. Some explanation can be also find in Discussion.

In our current study, the emphasis was the comparison of these methods using corresponding samples including plasma/tissue samples and we made great efforts to obtain comparable results. The purpose was not to present only results of the most suitable or appropriate method for OTA determination, but we aspired methods comparison and illustration of our findings.

  1. The authors mentioned in section 5 “On the other side, basic toxicological and morphological examinations were performed on animals or their excised organs throughout and/or at the endpoint of the experiment”, however, there are no related morphological results in the results.

Author response: Thank you for your perception. If we added additional images to our manuscript, it would be truly more informative. However, our team has previously detailed the morphological attributes and peculiarities of similar OTA exposure and those results have been already published. Plenty of palpable research papers exist on this topic.

In our recent paper, morphological changes/examinations were not our principal focus (moreover, based on exposure we expected no toxicological effects), because we primarily aimed demonstration of detection of low levels of mycotoxin and the exact comparison of three different techniques (HPLC, ELISA, and FCM). These observations were just supplementary experiments to further support the primary goal of the study. So, we would insist to keep the original “Results/Figures” part and images, but we modified the sentence to avoid misunderstandings.

  1. Figure 1. It seems no significance results were observed in the figure.

Author response: Yes, it is true. Based on our experiments there were no significant modifications in body weight as well as in organs weight between VC groups and OTA-exposed groups. These observations are signed in Figure 1. with ns (not significant). Therefore, the title of Figure 1. may cause some misconceptions, we use now “alterations” in title except of “changes”.

  1. Figure 2. OTA was detected in Vc group by FCM, please explain. Is it from the feed or others?

Author response: It could be also from feed (basal level of OTA in foodstuff has been measured, indicted in subsection 5.2) but also from the current assays. Our aim was not to build and validate “final” assays, some matrix effects or other uncertainties might still occur. Using such assays, we can detect mainly changes/differences following exposure. Validation based on standards might be an important task for future if it will be required

  1. Figure 3. For methods comparison, it seems correlation coefficient (r value) is low (less than 0.8). The authors should specify the most suitable method with high reliability.

Author response: Based on recent literature data, the interpretation of r-values are as follows: between 0.9 and 1.0 indicates a strong, positive correlation, between 0.8-0.9 denotes a fairly strong correlation, between 0.7-0.8 represents a significant, positive correlation, between 0.6-0.7 indicates a moderate, positive correlation and between 0.3 0.5 signifies low correlation, etc. Those were used also inter alia by GraphPad Prism v9.3.1. statistical software.

We are aware that in some cases r-values are lower than 0.8, (but they are above 0.7), which means a positive and significant relationship, not a low relationship. According to our results, the reliability of this value depends not only on the utilized method but also on the examined organs. All assays provided comparable results with strong/significant and positive correlations. Complexity, preferences as well as drawbacks of used assays are mainly detailed in Discussion.

The authors should specify the most suitable method with high reliability.

Furthermore, it is earlier answered above (question 2)

  1. Since this work is a methods comparison, the authors should explain more about the method validation in the results section.

Author response: As we have mentioned, validation based on standard procedures was not our current aim. HPLC methodology was earlier published as indicated. We applied spiking and calculated recovery of spiking/spiked samples in all measurements with HPLC, ELISA and FCM. Procedures which have been furtherly modified (MycoFossTM, TOXI-WATCH ELISA KIT) have been accordingly validated on original matrices (corn, wheat or barley) as indicated also by manufacturers. Reference samples related (organs, blood) are not available.

Reviewer 2 Report

Abstract

Indicate which method can detect OTA at the lowest level. Compare values of detected OTA between blood, kidney and liver.

L6-7: rephrase

Introduction

L58-59: indicate also reference for these investigations

L112: indicate these tissues

Results

Why liver weight was not determined and why concentration of OTA was not determined in spleen?

L120: why 3-day exposure was selected? This time can be too short to determine significant differences in body or organ weight.

L199, 201: what is SI?

Discussion

L281: add more references for this statement (e.g. Lozowicka et al. Impact of Diversified Chemical and Biostimulator Protection on Yield, Health Status, Mycotoxin Level, and Economic Profitability in Spring Wheat (Triticum aestivum L.) Cultivation. Agronomy 2022, 12, 258. https://doi.org/10.3390/agronomy12020258)

Conclusions

Conclude which method is the most sensitive (detection of OTA at the lowest level) and in which tissues/blood the OTA level was the highest/lowest.

Materials and Methods

L426: -20 °C

L436: dry meat?

L482, 487: precisely what eluent do you mean?

L483: indicate mass of kidney and liver

L493: it is a precolumn. Remove the second word ‘guard’ in this line

L521: why the range was indicated for constant values?

L548: indicate the volume of washing buffer per well

L549: do not start the sentence with a number

L550: add a volume of sulfuric acid

L590: ‘washed out’ – volume

L595-596: describe these abbreviations

Author Response

Referee report from Reviewer 2

Major comments:

  1. Abstract

Indicate which method can detect OTA at the lowest level. Compare values of detected OTA between blood, kidney and liver.

Author response: It has been now indicated. However, we had to rephrase whole abstract because of 200 words limitation.

However, we think main importance is to demonstrate dedicated assays (not only HPLC) could be set and applied for measurements of OTA in tissues and blood. Assays should be validated accordingly to standards, currently some matrix effect or other uncertainties might still occur.

L6-7: rephrase

Author response: It has been now rephrased.

  1. Introduction

L58-59: indicate also reference for these investigations

Author response: It has been now indicated.

L112: indicate these tissues

Author response: It has been now indicated, kidney and liver are now mentioned.

We have attempted to correct the manuscript (Introduction) according to your comments.

  1. Results

Why liver weight was not determined and why concentration of OTA was not determined in spleen?

Author response: As we mentioned our main aim was to detect (develop appropriate assays for this purpose) presence or accumulation of OTA following exposure to low level. We have excised main organs and some of them have been processed/subjected to measurements with assays.

The reason why OTA was estimated from kidney and liver is additionally that in future studies we would like to routinely determine mycotoxins in these organs of domestic, and wild animals. According to the literature, the mass of the spleen may increase upon OTA exposure and we took these observations as a basis, so spleen samples were also measured. Most probably due to the short exposure periods, we did not notice remarkable increments in the spleen mass of exposed animals compared to the VC group (please see Figure 1.).

L120: why 3-day exposure was selected? This time can be too short to determine significant differences in body or organ weight.

Author response: We have chosen 3-day exposures because in our earlier experiments we also applied these/similar periods. We agree with the reviewer on the comment that this exposure period might be too short to assess significant alterations in body or organs weight, but the principal goal was mainly to determine presence, accumulation of OTA even following short exposure to small concentration.

L199, 201: what is SI?

Author response: SI means the supplementary information. It is only the abbreviated form and has been signed in the manuscript.

  1. Discussion

L281: add more references for this statement (e.g. Lozowicka et al. Impact of Diversified Chemical and Biostimulator Protection on Yield, Health Status, Mycotoxin Level, and Economic Profitability in Spring Wheat (Triticum aestivum L.) Cultivation. Agronomy 202212, 258. https://doi.org/10.3390/agronomy12020258)

Author response: Additional reference has been added as requested.

  1. Conclusions

Conclude which method is the most sensitive (detection of OTA at the lowest level) and in which tissues/blood the OTA level was the highest/lowest.

Author response: It was also requested above (Assays perform similarly, but overwhelmingly HPLC measured at lowest exposure to OTA. Additionally, blood contained the highest level of OTA, mycotoxin accumulation in organs examined was lower, as expected). Now it is concluded.

  1. Materials and Methods

L426: -20 °C

Author response: It was corrected in the manuscript based on the suggestion of the Reviewer 2.

L436: dry meat?

Author response: Dry meat was corrected to dry matter in the manuscript

L482, 487: precisely what eluent do you mean?

Author response: The HPLC eluent applied for the HPLC-FLD analyses: sodium borate buffer (10 mM, pH 10.0) and ACN (83:17 v/v%). To make it clear, it has also been inserted into line 482. However, according to question from reviewer 3, the following sentence has been used: (sodium borate buffer (10 mM, pH 10.0) and ACN/water solution (83/17 v/v)

L483: indicate mass of kidney and liver

Author response: It has been now indicated. Kidney data are given in supplementary file (Table S1), mass of whole liver has not been measured because of technical/practical issues and its low importance.

L493: it is a precolumn.

Author response: Guard column was changed to precolumn based on the suggestion of the Reviewer.

L521: why the range was indicated for constant values?

Author response: Authors do not understand the question, we do not find related issue at given line. „Range” was given for ELISA Kit (originally at L536), represented the standards found in Kit.

L548: indicate the volume of washing buffer per well

Author response: It was corrected/ supplemented in the manuscript based on the suggestion of the Reviewer 2.

L549: do not start the sentence with a number

Author response: It was corrected in the manuscript based on the suggestion of the Reviewer 2.

L550: add a volume of sulfuric acid

Author response: It was corrected/supplemented in the manuscript based on the suggestion of the Reviewer 2.

L590: ‘washed out’ – volume

Author response: It was corrected/supplemented in the manuscript based on the suggestion of the Reviewer 2.

L595-596: describe these abbreviations

Author response: Thank you for your notifications. According to your suggestions we performed the corrections in the manuscript

Reviewer 3 Report

The article “Analysis and Comparison of Rapid Methods for the Determination of Ochratoxin A Levels in Organs and Body Fluids Obtained from Exposed Mice” is focused on developing analytical solutions for Ochratoxin A (OTA) detection/quantification at low concentrations in animal plasma and organs. The research background is sufficiently described, referring to the persistent organic pollutants (POPs) such as OTA being able to interfere with the physiology and/or behaviour of exposed organisms, and pointing out the importance of the availability of analytical methods for accurate OTA (and POPs in general) determination in complex matrices. As a result of their research, dedicated assays on different technical platforms (HPLC, ELISA, FCM) have been developed, tested using same samples and providing comparable results.

The methods used in this research are adequately described, and the results are well presented and discussed. The reference list is substantial, not entirely current but relevant, while English language and style are generally satisfying. There are also few other minor observations to be addressed before publication.

Lines 10-11 – I would suggest to write either full name or abbreviations for all three techniques.

Line 17 – Consider losing might from the sentence We might demonstrate that … or modify it differently.

Lines 105-107 – I would suggest rewriting this sentence, it sounds like you’re diminishing your work by using at least... You could perhaps write something like: Since assay development tends to be very complex and time-consuming process, at this point we focused on finding and proving applicability and adaptability of different techniques for the afore-mentioned purpose.

Line 412-419 – In my opinion should be rewritten, I suggest you to lose We might conclude that, as a demonstration from the first sentence, and might from line 417.

Line 478 – Perhaps to write HPLC-FLD was used as a base, instead of as basic.

Line 526 – Instead of writing 84 % (v/v) ACN:water solution, I would suggest you to write ACN/water solution (84/16, v/v). Apply where applicable.

Line 552 – The recovery was 77.6-65.42 % – Do you mean 65.42-77.6%? Please check and rewrite. Also, it would be advisable to unify the number of decimal places.

Author Response

Referee report from Reviewer 3

General comments:

The article “Analysis and Comparison of Rapid Methods for the Determination of Ochratoxin A Levels in Organs and Body Fluids Obtained from Exposed Mice” is focused on developing analytical solutions for Ochratoxin A (OTA) detection/quantification at low concentrations in animal plasma and organs. The research background is sufficiently described, referring to the persistent organic pollutants (POPs) such as OTA being able to interfere with the physiology and/or behaviour of exposed organisms, and pointing out the importance of the availability of analytical methods for accurate OTA (and POPs in general) determination in complex matrices. As a result of their research, dedicated assays on different technical platforms (HPLC, ELISA, FCM) have been developed, tested using same samples and providing comparable results.

The methods used in this research are adequately described, and the results are well presented and discussed. The reference list is substantial, not entirely current but relevant, while English language and style are generally satisfying. There are also few other minor observations to be addressed before publication.

Minor comments:

  1. Lines 10-11 – I would suggest to write either full name or abbreviations for all three techniques.

Author response: Thank you for suggestion, the names for all three techniques are modified accordingly. However, because of questions from Reviewer 2 and the 200 words limitation whole abstract has been rephrased.

  1. Line 17 – Consider losing might from the sentence We might demonstrate that … or modify it differently.

Author response: Thank you for suggestion, we considered and modified the sentence, accordingly.

  1. Lines 105-107 – I would suggest rewriting this sentence, it sounds like you’re diminishing your work by using at least... You could perhaps write something like: Since assay development tends to be very complex and time-consuming process, at this point we focused on finding and proving applicability and adaptability of different techniques for the afore-mentioned purpose.

Author response: Thank you for your proposal, it is a decent point that we need to consider. The sentence has been corrected and it is clarified now in the manuscript according to your comment.

  1. Line 412-419 – In my opinion should be rewritten, I suggest you to lose We might conclude that, as a demonstration from the first sentence, and might from line 417.

Author response: Thank you for your notification. We agree with the reviewer comment, and these sentences in (4.) Conclusion part have been modified, accordingly. Thus, our conclusion contains exact and precise statements, instead of ambiguous arguments.

  1. Line 478 – Perhaps to write HPLC-FLD was used as a base, instead of as basic.

Author response: Thank you for your remark, we use “as a base” instead of “as a basic” according to your comment.

  1. Line 526 – Instead of writing 84 % (v/v) ACN:water solution, I would suggest you to write ACN/water solution (84/16, v/v). Apply where applicable.

Author response: The “84 % (v/v) ACN:water solution” were corrected to “ACN/water solution (84/16, v/v)” according to your comment. These modifications can be found in Line 497, 519 and 528. (similar request occurred from Reviewer 2 which was also considered)

Line 552 – The recovery was 77.6-65.42 % – Do you mean 65.42-77.6%? Please check and rewrite. Also, it would be advisable to unify the number of decimal places.

Author response: Thank you for your remark, it is clarified (65.4-77.6) now in the manuscript.

Round 2

Reviewer 1 Report

All the comments were responsed and revised. I think this work could be published in the journal TOXINS

Author Response

Referee report from Reviewer 1

Comments and Suggestions for Authors

All the comments were responsed and revised. I think this work could be published in the journal TOXINS

Author response: We would like to thank the Reviewer for help and comments needed for improving our manuscript.

As authors can understand, “Extensive editing of English language and style” is “only” required (now). In previous revision “Moderate English changes” were required by Reviewer.

Considering inputs from the other Reviewers, we would like to have/obtain help, opinion, and suggestion of Editors. If language correction is needed, please, provide us related quotation/information on service arranged by MDPI.

Reviewer 2 Report

The Authors have improved the manuscript, however, in Abstract the values of OTA concentration between blood and tissues should be compared as previously indicated.

L528: why 0.25-0.25 g and 0.5-0.5 ml was indicated instead 0.25 g and 0.5 ml?

Author Response

Referee report from Reviewer 2

Comments and Suggestions for Authors

The Authors have improved the manuscript, however, in Abstract the values of OTA concentration between blood and tissues should be compared as previously indicated.

L528: why 0.25-0.25 g and 0.5-0.5 ml was indicated instead 0.25 g and 0.5 ml?

In Abstract the values of OTA concentration between blood and tissues should be compared as previously indicated.

Author response: Abstract has been modified as (hopefully) required.

In previous revision the following related comment has been raised: “Compare values of detected OTA between blood, kidney and liver.” As an answer, we then added “Blood contained the highest level of OTA, accumulation in organs was lower”. Now, that correction is replaced to “Blood contained higher level of OTA than liver and kidney.”.

We hope that we can now understand this request from Reviewer. However, we would like to also give a short explanation.

Following exposure, blood contained higher level of OTA (usually about tenfold concentration!) than liver or kidney measuring with HPLC, ELISA or FCM.

Level of OTA accumulated in kidney was similar with the OTA concentration found in liver. However, we have to say these assays are not “final”, validated methods and they use different extraction procedures as well. Our aim was mainly to demonstrate, dedicate and practical assays can (and should) be developed and applied.

Additionally, considering our aim and the 200 words limitation (abstract) we would avoid to give/report many of values (numbers) in Abstract.

L528: why 0.25-0.25 g and 0.5-0.5 ml was indicated instead 0.25 g and 0.5 ml?

Author response: We can agree and apologize. It has now been modified to 0.25 g and 0.5 ml.

As an explanation, it might be because of “Hungarian use/phrase for the meaning of 0.25 g (or 0.5 ml) per each sample (applied on each sample).

And finally, we would like to thank the Reviewer for help and comments needed for improving our manuscript.
